# Effects of High Irradiance and Low Water Temperature on Photoinhibition and Repair of Photosystems in Marimo (*Aegagropila linnaei*) in Lake Akan, Japan

**DOI:** 10.3390/ijms24010060

**Published:** 2022-12-21

**Authors:** Akina Obara, Mari Ogawa, Yoichi Oyama, Yoshihiro Suzuki, Masaru Kono

**Affiliations:** 1Department of Biological Sciences, Faculty of Science, Kanagawa University, 2946 Tsuchiya, Kanagawa, Hiratsuka 259-1293, Japan; 2Department of Primary Education, Faculty of Education, Yasuda Women’s University, 6-13-1 Yasuhigashi, Asaminami-ku, Hiroshima 731-0153, Japan; 3Marimo Research Center, Kushiro Board of Education, Hokkaido, Kushiro 085-0467, Japan; 4Department of Biological Sciences, Graduate School of Science, The University of Tokyo, 7-3-1 Hongo, Bunkyo-Ku, Tokyo 113-0033, Japan

**Keywords:** cell death, chlorophyll fluorescence, F_v_/F_m_, Lake Akan, low temperature, marimo, PAM, photoinhibition, repair

## Abstract

The green alga *Aegagropila linnaei* often forms spherical aggregates called “marimo” in Lake Akan in Japan. In winter, marimo are exposed to low water temperatures at 1–4 °C but protected from strong sunlight by ice coverage, which may disappear due to global warming. In this study, photoinhibition in marimo was examined at 2 °C using chlorophyll fluorescence and 830 nm absorption. Filamentous cells of *A. linnaei* dissected from marimo were exposed to strong light at 2 °C. Photosystem II (PSII) was markedly photoinhibited, while photosystem I was unaffected. When the cells with PSII damaged by the 4 h treatment were subsequently illuminated with moderate repair light at 2 °C, the maximal efficiency of PSII was recovered to the level before photoinhibition. However, after the longer photoinhibitory treatments, PSII efficiency did not recover by the repair light. When the cells were exposed to simulated diurnal light for 12 h per day, which was more ecological, the cells died within a few days. Our results showed new findings of the PSII repair at 2 °C and serious damage at the cellular level from prolonged high-light treatments. Further, we provided a clue to what may happen to marimo in Lake Akan in the near future.

## 1. Introduction

*Aegagropila linnaei*, a unique freshwater green alga of the order Cladophorales, consists of 1–4 cm long filamentous cells and is distributed not only in several lakes in Japan but also in many lakes of some regions in Europe, East Asia, and North America [1,2]. This species grows as attached, floating, and unattached forms. Interestingly, *A. linnaei* also forms spherical aggregates known as “lake balls” [1], “Cladophora balls” [3], or “marimo”, in which “mari” means “ball” and “mo” means “alga” in Japanese [2]. Lake Akan in Hokkaido, Japan (Figure 1a), is the only lake in which spherical marimo is found in Japan. Marimo grown in Lake Akan could become as big as 30 cm in diameter, and such large marimo have not been reported from anywhere in the world. Unfortunately, the abundance of marimo is decreasing at the world scale due to global anthropogenic changes, and as a result, the large colonies are observed only in this lake now (Figure 1b). With an aim of conservation, marimo in Lake Akan was designated as a Natural Monument of Japan in 1921 and a Special Natural Monument in 1952. Marimo in Lake Akan have experienced dynamic changes in water temperatures throughout the year. In summer, water temperatures are 20–27 °C (data not shown). In winter, water temperatures lower to 1–4 °C (Figure 1e), and the lake surface is covered with about 50 cm thick ice with accumulated snow in winter (Japan Meteorological Agency; https://www.jma.go.jp/jma/indexe.html (1 November 2022)). The ice and snow coverage on the lake decrease irradiance reaching to the bottom of the lake and protect marimo against strong sunlight. However, the ice and snow coverage may disappear due to global warming in the near future, and then marimo could be exposed to strong irradiance at low temperatures.

The photosynthetic electron transport is driven by photochemical reactions of two photosystems, PSII and PSI, in the thylakoid membrane, and the electrons finally reduce NADP^+^ to NADPH. The electron flow also generates the transmembrane electrochemical potential difference of H^+^ to drive the H^+^-ATP synthase to produce ATP. While light drives photosynthesis, it also damages the photosynthetic apparatus. Light in excess of the photosynthetic capacity, namely, light energy that is not used in photochemistry or dissipated safely as heat, would lead to the formation of reactive oxygen species (ROS), which attack the two photosystems, PSII and PSI [4]. Such damage could lead to the loss of photosynthetic activity, so-called photoinhibition. PSII is sensitive to a variety of environmental factors such as high light and low temperatures [5,6,7,8]. While PSII is vulnerable to photodamage, the damaged PSII needs to be rapidly repaired: the damaged PSII reaction center protein is degraded [9,10], the new one is synthesized, and PSII is subsequently re-activated [11,12]. The repair process needs light and proceeds on the order of minutes under non-stress conditions [13]. In moderate light, the PSII damage and repair are balanced. When the rate of the damaging is greater than that of the repair, the loss of photosynthetic activity, or photoinhibition, becomes apparent. Thus, the extent of photoinhibition in vivo depends on a balance between the photodamage and the recovery from the photoinhibited state. High-light sensitivity of PSII is considerably enhanced when plants and algae are exposed to low temperatures because chilling-light exposure could accelerate the photodamage to PSII [14,15] and/or reduce the rate of PSII repair [16,17,18]. In contrast to PSII, which is highly susceptible to photodamage, PSI is quite resistant to stress from stable strong light. Photoinhibition of PSI occurs under characteristic conditions such as moderate light at chilling temperatures [19,20] and fluctuating light [21,22]. Once PSI inhibition occurs, it strongly suppresses photosynthesis [23] as its repair and/or de novo synthesis are very slow [24].

Algae and plants have evolved some photoprotective mechanisms to avoid these photoinhibitions. Excess absorbed light energy is dissipated in the PSII antenna complex as heat, which can be measured fluorometrically as the non-photochemical quenching (NPQ). NPQ is an efficient process that protects PSII reaction centers from photoinhibition, and the light-induced NPQ is mainly regulated by the proton gradient across the thylakoid membrane [25,26]. For PSI, keeping the fraction of oxidized PSI reaction center P700, P700^+^, is essential to protect PSI from photoinhibition [27,28]. Cyclic electron transport around PSI [22,29,30], photosynthetic control at the cytochrome *b_6_/f* complex [31,32,33], and PSII photoinhibition [34] contribute to PSI protection.

For the conservation of marimo, it is necessary to understand photoinhibition in marimo at low temperatures and whether the repair of the damaged PSII occurs even at low temperatures. However, photosynthetic properties in marimo have not been studied intensively [35,36], especially at low temperatures in winter. In this study, we examined photoinhibition of PSII and PSI in marimo against exposure to high light at 2 °C, and recovery at 2 °C from the photoinhibited state, in marimo grown in Lake Akan in winter.

## 2. Results

### 2.1. Environmental Conditions at the Sampling Site of Churui Bay in Lake Akan

During the winter in 2021 and 2022, the water surface at the sampling site of Churui Bay in Lake Akan (Figure 1a) was covered with ice about 50 cm thick (Figure 1c). Water temperatures just beneath the ice were 0–1 °C and gradually increased to ca. 2.0 °C toward the bottom at 1.5 m in depth (red line in Figure 1e). Photosynthetic photon flux density (PPFD) on the ice was high at 1500–2000 μmol m^−2^ s^−1^ at around noon on sunny days. PPFD on the bottom of the lake inhabited by marimo was less than 10 μmol m^−2^ s^−1^. In the light spectrum measured at the bottom, a green waveband of 500–600 nm was more abundant than the red and blue bands of 400–500 and 600–700 nm (Appendix A). To estimate the environmental conditions without ice coverage in the future, we cut the ice at a size of 2.5 × 2.5 m (Figure 1d). Soon after the ice removal, the vertical profile of temperatures was similar to that with the ice (black line in Figure 1e). The water surface was frozen within a day, the ice thickness increased for a few days, and the water temperature at the bottom was maintained at ca. 2.0 °C. Through the ice hole, strong sunlight at relatively high PPFDs of 500–800 μmol m^−2^ s^−1^ reached the bottom of the lake (Figure 1f). Absorption of lake water was not strong. The light spectrum just under the water without the ice was similar to that of sunlight on the lake (Appendix A).

### 2.2. Measurements of F_v_/F_m_ and P_m_ against the Exposure to Light-Chilling Treatments

We examined the photoinhibition against strong light at 2 °C in marimo. Marimo samples were collected at Lake Akan and stored at 2 °C in the dark. Filamentous cells of *A. linnaei* were dissected from the surface of the marimo and were used for measurements. Experiments were carried out at 2–4 °C in a cold room. The *A. linnaei* filamentous cells were illuminated by continuous high light from white LEDs at a PPFD of 800 μmol m^−2^ s^−1^ for up to 180 min (Figure 2). The maximum quantum yield of PSII photochemistry, F_v_/F_m_, before the high-light treatment was ca. 0.45. This value was lower compared with those in land plants (ca. 0.78–0.84) and in algae (ca. 0.5–0.6). One of the potential reasons for low F_v_/F_m_ was that the pool of plastoquinone (PQ), a mobile electron carrier, between the PSII and cytochrome *b_6_/f* complex in the thylakoid membrane, might be somewhat reduced in the dark. To examine the effect of the reduced PQ pools on the minimum level of Chl fluorescence in the dark, F_o_, we applied the far-red light in the dark to oxidize the PQ pool by preferentially exciting the PSI reaction center, P700 (Appendix A). While P700^+^ was increased either by the far-red pulse or continuous illumination of far-red light, Chl fluorescence never responded. This indicated that the F_o_ level was correctly measured. In another confirmatory experiment, we found that P700 signal in the dark before the SP was slightly oxidized, indicating that the PQ pool would be also oxidized in the dark (Appendix A).

The *A. linnaei* filamentous cells were illuminated to continuous high light with the white LEDs at a PPFD of 800 μmol m^−2^ s^−1^ for up to 180 min (Figure 2). F_v_/F_m_ of ca. 0.45 before the high light slightly decreased by 15 min and markedly decreased to ca. 0.2 after exposure for 30 min. The F_v_/F_m_ levels showed similar values for up to 180 min (black bar in Figure 2a). In contrast, the maximum oxidizable P700 contents, P_m_, did not change from the exposure to high light, indicating that no PSI was photoinhibited (Figure 2b). Next, to examine the repair activity from the photoinhibited state, we illuminated the PSII-photoinhibited filamentous cells with a moderate light of 250 μmol m^−2^ s^−1^, “repair light,” for 30 min at 2 °C. F_v_/F_m_ levels in all the PSII-photoinhibited filamentous cells were remarkably recovered to the level before the damage even when the repair treatment was conducted at 2 °C (orange bar in Figure 2a).

### 2.3. Quantum Yields of PSII and PSI of the PSII in the Photoinhibited Cells

The repairing process of photoinhibited PSII was analyzed with the three states of cells: the intact cells, the cells photoinhibited by strong illumination for 60 min (the inhibited cells), and the cells repaired by illuminating the inhibited cells with the repair light at 250 μmol m^−2^ s^−1^ for 30 min (the repaired cells). The quantum yields of the PSII photochemistry, Y(II), and light-induced excess-energy dissipation, Y(NPQ), and the fraction of open PSII centers, qL, were estimated with the chlorophyll fluorescence. In addition to the fluorescence, the 830 nm absorption was also simultaneously measured. For indices of PSI, the quantum yields of the PSI photochemistry (Y(I)), the donor-side limitation of PSI (Y(ND)), and the acceptor-side limitation of PSI (Y(NA)) were determined. We can estimate the process of inhibition from the differences in the light response of these parameters among the cells.

The redox states of PSI showed no significant differences by PSII photoinhibition: Y(I) was decreased with the PPFD levels, Y(ND) was increased with the PPFD levels, and Y(NA) reached the maximum at a moderate PPFD of 250 μmol m^−2^ s^−1^ (Figure 3a,c,e). Y(II) in the inhibited cells was markedly reduced at low to moderate PPFDs up to 250 μmol m^−2^ s^−1^. Y(II) in the repaired cells was similar to that in the intact cells (Figure 3b). The inhibited cells showed the largely depressed Y(NPQ) (Figure 3d) and slightly higher qL than the intact and repaired cells at PPFDs above 250 μmol m^−2^ s^−1^ (Figure 3f).

### 2.4. Photoinhibition under the Expected Natural Environment without the Ice in Winter

We assumed that the water temperature would become low even without the ice on the lake. To examine the photosynthetic responses under more ecological conditions, we adopted the diurnally changing illumination. For the control, we also used continuous illumination for 12 h. First, F_v_/F_m_ in the filamentous cells of *A. linnaei* was analyzed with the exposure to continuous photoinhibitory light at 500 μmol m^−2^ s^−1^ for 12 h (Figure 4a). The F_v_/F_m_ of ca. 0.45 before the treatment was decreased to 0.25 after the photoinhibitory light for 2 h, and the same level was kept until the end of the experiment (Figure 4b). The illuminated cells were treated in the dark for 12 h (black bars in Figure 4c), followed by the repair light treatment at 250 μmol m^−2^ s^−1^ for 30 min. F_v_/F_m_ in the photoinhibited cells for 2 h and 4 h recovered to levels similar to those of the non-damaged cells, but F_v_/F_m_ in the cells photoinhibited for more than 6 h showed no recovery at all (orange bar in Figure 4c).

Next, we used a diurnal condition as the more realistic pattern. The filamentous cells were exposed to the simulated natural illumination for a 12 h light period in a day: The PPFD level increased gradually from 0 μmol m^−2^ s^−1^ to 500 μmol m^−2^ s^−1^ at 6 h and decreased gradually to 0 μmol m^−2^ s^−1^ at 12 h (Figure 5a). When the PPFD reached 250 μmol m^−2^ s^−1^ after 3 h from the onset, F_v_/F_m_ was not decreased. At 6 h from the onset of the illumination, F_v_/F_m_ markedly decreased to ca. 0.2 (Figure 5b). After 6 h from the onset of the illumination, the cells were exposed to the illumination at less than 500 μmol m^−2^ s^−1^ again. However, F_v_/F_m_ remained consistently low until the end of the illumination. Even when treated with the repair light for 30 min, F_v_/F_m_ in these photoinhibited cells did not recover (Figure 5c).

After 2 days of the simulated natural illumination, F_v_/F_m_ showed a lower value of ca. 0.1, and it did not change until 5 days (Figure 6). We examined whether the photoinhibited cells after 5 days of illumination recovered with a modulated repair light for 12 h in a day, in which PPFD changed with half the intensity of the simulated natural illumination. Even when repair light/dark cycles were repeated for 14 days, no recovery of F_v_/F_m_ was observed, and F_v_/F_m_ was further decreased during this period (Figure 6). The death of the cells was confirmed by the neutral red stain, although the cells kept their green colors even after 15 days at the low temperature (Figure 7b–d, and also refer to Appendix A).

## 3. Discussion

Churui Bay in Lake Akan, the habitat of the special natural treasure of marimo, is covered with ice reaching 50 cm thickness in winter. There is also much snow on top of the ice coverage, and thus, light intensity is extremely low at about 10–40 μmol m^−2^ s^−1^ in the marimo habitat (Figure 1 and Appendix A). However, the ice thickness has been progressively thinner in recent years due to global warming. When the ice coverage was removed, the light intensity reaching the lake bottom increased to 500–800 μmol m^−2^ s^−1^ (Figure 1f). These areas of open water were covered by ice more than 10 cm thick within a few days. Since the ice acted as a heat shield, water temperatures could maintain 0–4 °C even though ice thickness became thinner or partly collapsed. Under these conditions without ice coverage, marimo could be exposed to strong light at low temperatures during the winter for a few months. This is thought to be dangerous for photosynthesis and the survival of marimo, while in warm water, such as in summer, both photoinhibition and cell death could be avoided (Appendix A). In this study, we examined the response of the green alga *A. linnaei*, which forms the marimo lake balls, to the anticipated low temperatures and high light intensity in the future.

For land plants, PSI is reported to be sensitive against the chilling-light exposure in chilling-sensitive plants such as cucumber [19,37], coffee [38], cotton [14], and common bean (*Phaseolus vulgaris* L.) [39] and even in chilling-tolerant plants such as *Arabidopsis thaliana* [40]. Fluctuating light is also a potent stress factor for PSI. PSI photoinhibition by the fluctuating light has been reported in many species including *A. thaliana* [21,22,41], rice [42], sunflower [43], and field-grown plants [27]. On the other hand, although photoinhibition of PSI by fluctuating light was reported in cyanobacteria [44,45] and the green alga *Chlamydomonas reinhardtii* [46,47], there is little information about PSI sensitivity to low temperatures in algae [7,17,48]. In the present study, PSI in marimo acclimated to low temperatures was not affected by the strong light (Figure 2 and Figure 3), although PSI photoinhibition by fluctuating light was not examined.

In contrast to PSI, PSII was markedly photoinhibited in the high light even for 30 min at 2 °C (Figure 2), while no PSII photoinhibition occurred at 20 °C (Appendix A). F_v_/F_m_ in the dark-adapted cells was lower than those reported for other algae. Although the reduced PQ pools were unlikely to affect the F_o_ and F_m_ in the dark (Appendix A), sustained thermal dissipation might have been induced in marimo grown in the low temperatures in Lake Akan.

In response to the actinic light, qL was higher and Y(II) was lower in the inhibited cells compared with the control (Figure 3). These results indicated that the electron flow heading to PSI was suppressed due to PSII damage and the PQ pool was more oxidized in the inhibited cells than the other cells. The proton gradient across the thylakoid membrane could have been small in the inhibited cells, resulting in the higher qL. The lower efficiency of PSII could have avoided the damage to PSI induced by the stress [20,25]. Further, inhibited marimo showed low Y(NPQ) (Figure 3). This parameter reflects light-induced NPQ, namely, pH-dependent energy quenching [49,50]. Cyclic electron transport around PSI also contributes to the formation of the proton gradient across the thylakoid membrane. Low Y(NPQ) might mean that the cyclic pathway contributed less to the PSI protection. From these results, we concluded that PSII photoinhibition mainly protected PSI.

The repair process of PSII has been proposed to follow the degradation of the damaged PSII reaction center protein and subsequent de novo synthesis [9,10,11,12]. The repair of damaged PSII is believed to only occur at fairly slow rates at low temperatures in photosynthetic organisms living at room temperatures [17]. We have demonstrated that damaged PSII in filamentous cells is repaired under moderate light at low temperatures of 2 °C but not in the dark. These results suggest that the filamentous cells have a PSII photo-repair system that maintains activity at low temperatures [51,52,53]. In the present study, we could not conduct detailed experiments such as light-/time-dependent properties of the PSII repair and the repair activity at room temperatures, because the sample size was limited because marimo are a Special Natural Monument. These are important issues to consider in future work.

We further showed that PSII in marimo cells showed irreparable photoinhibition after 6 h of the simulated illumination ranging from 250 to 500 μmol m^−2^ s^−1^, while PSII photoinhibition occurring for 4 h of the light treatment was repairable (Figure 4, Figure 5 and Figure 6). These results suggested that relatively long exposure of the cells to strong light might have exceeded a certain threshold level of a balance between the damage and repair of PSII by suppressing the repair activity or accelerating the damage to PSII, causing severe damages, which subsequently led to cell death (Figure 7). The thermal energy dissipation of absorbed light energy is thought to act to avoid ROS-mediated inhibition of the de novo synthesis of the PSII reaction center protein [54]. Y(NPQ) in the damaged filamentous cells in this study was drastically low in whole range of the PPFD levels (Figure 3). Low capacity of NPQ under the photoinhibitory condition might cause the suppression of the PSII repair by allowing ROS to accumulate within the chloroplast [4].

In the present study, as we used the dissected filamentous cells, the results might be only valid at the cellular level. In fact, we did not consider the effects of the morphological structure of the spherical marimo on the protection of PSII against the exposure to high illumination. Nonetheless, our results provide an indicator or clue to what may happen to the spherical marimo in Lake Akan. The fact of the cell death caused by the photoinhibitory treatment for only 6 h at 2 °C suggests that photoinhibition would be a serious threat to the surface part of marimo in the lake when global warming proceeds. The natural habitat of marimo receives more than 10 h of sunlight, even in winter. If the damage of the surface cells increases under the longer exposure to direct sunlight per day due to the thinning of ice or ice collapse, in an extreme case, this may affect maintenance of their round bodies and lead to the disappearance of giant marimo. There is another possibility. It is known that phytoplankton in lake water increases when the lake surface is unfrozen. This suggests a greater absorption of sunlight by the lake water, and it might alleviate photoinhibition in marimo at low temperatures. Nonetheless, the irreversible photoinhibition of PSII in the filamentous cells indicated by this study suggests that this may cause survival of marimo to be more difficult in the near future. If global warming proceeds, there must be a stage where ice will melt away to allow high light penetration to the lake bottom while the water temperature is still low. We need to monitor the environmental conditions in Lake Akan continuously and to examine characteristics of the photoinhibition and repair at low temperatures in the spherical marimo themselves. Furthermore, we must urgently deal with protecting the marimo habitat.

## 4. Materials and Methods

### 4.1. Sites and Sample Collection

Spherical marimo samples of *A. linnaei* were collected at Churui Bay in Lake Akan (center: 43°27′ N, 144°06′ E, boundary length: 30.3 km, surface area: 13.3 km^2^, and mean depth: 17.8 m), Hokkaido, Japan, on 9 March 2022 (Figure 1a). Natural marimo samples, the size of which was about 10–15 cm in diameter (*n* = 3), were collected by hand. They were immediately transferred to a water bath filled with lake water at 0–2 °C and stored in the dark until measurements were taken. When taking the measurements, we dissected the filamentous cells from the surface part of the spherical marimo with tweezers.

### 4.2. Vertical Profile of Water Temperature and Light Intensity under Ice and After Ice-off in the Lake

For the environmental measurements, we chose a site in the area of Lake Akan located 80 m away from the shore in Churui Bay (about 1.5 m depth; Figure 1a). For protective reason, these were carried out near the habitat of marimo. We measured the vertical profile of environmental factors under the water. We bored a small hole in the ice with a drill and settled a CTD probe (RINKO-Profiler, JFE Advantech Co., Ltd., Nishinomiya, Japan) or a PPFD sensor (DEFI2-L, JFE advantech Co., Ltd., Nishinomiya, Japan) through the hole. These were referred to as “data under the water with ice.” When data under the water after ice-off were measured, we cut out the ice at a size of about 2.5 × 2.5 m, and the same measurements were carried out as above.

The spectra of the sunlight under the water were also measured with a handheld portable Light Analyzer LA-105 (Nippon Medical & Chemical Instruments Co., Ltd., Osaka, Japan).

### 4.3. Photoinhibitory and Repair Treatments

All handling and treatments were carried out in a cold room at 4 °C. A square array of 36 white LEDs peaked at 591 nm and 449 nm covered with a transparent plastic plate (15 cm × 15 cm, laboratory handmade, for the spectra; see Appendix A) was used. The filamentous cells of *A. linnaei* were carefully picked up from the surface part of the spherical marimo with tweezers (Appendix A) and placed on a 24-well plate, filled in lake water, on the ice in the dark for at least 30 min. Photoinhibitory light was provided to the filamentous cells from straight above at PPFD levels of 800 or 500 μmol m^−2^ s^−1^ at the sample surface, for different periods (Appendix A). At the given time, several filamentous cells were picked up from the well and used to measure the chlorophyll fluorescence and/or 830 nm absorbance. The remaining samples in the well continued to be illuminated for the next measurements. After the photoinhibitory treatments, the samples were dark-treated overnight at the low temperature. The next day, the white light used as the repair light at 250 μmol m^−2^ s^−1^ irradiated these photoinhibited samples for 30 min. Each data point was independently measured with different samples.

### 4.4. Chlorophyll Fluorescence and 830 nm Absorbance Change Measurements

For measurements with suspensions using a cuvette, chlorophyll fluorescence and absorption changes at 830 nm were measured simultaneously using a DUAL-PAM-100 (equipped with a fluorometer for chlorophyll fluorescence and a near-infrared detector for P700 absorption analysis with emitters at 830 and 875 nm, which were mounted on an optical unit ED-101US/MD; Walz, Effeltrich, Germany) in the cold room at 4 °C. According to [55,56,57,58], we determined the optimal settings for each parameter, such as intensities and widths of the measuring light and saturating light. Saturating pulses (SPs) from red LEDs (6000 μmol m^−2^ s^−1^, 500 ms duration) were applied to determine the maximum chlorophyll fluorescence with closed PSII centers after dark treatment for at least 30 min (F_m_) and during illumination (F_m_′). The maximum photochemical quantum yield of PSII (F_v_/F_m_, [59]) and the effective quantum yield of PSII (Y(II)) in the actinic light were calculated as (F_m_ − F_o_)/F_m_ and (F_m_′ − F_s_′)/F_m_′ [60], respectively, where Fs’ is the steady-state chlorophyll fluorescence level in the actinic light from red LEDs. Other PSII quantum yields, Y(NPQ) and Y(NO), which represent the regulated and non-regulated energy dissipation in PSII, respectively, were calculated as F_s_′/F_m_′—F_s_′/F_m_ and F_s_′/F_m_, respectively [61,62,63]. These add up to unity with the photochemical quantum yield (i.e., Y(II) + Y(NPQ) + Y(NO) = 1). The coefficient of photochemical quenching, qL, a measure of the fraction of open PSII reaction centers, based on the lake model of PSII antenna pigment organization, was calculated as (F_m_′ − F_s_′)/(F_m_′ − F_o_′) × F_o_′/F_s_′ [62]. F_o_′ is the minimal fluorescence yield in the actinic light and was estimated using the equation of [64], F_o_/(F_v_/F_m_ + F_o_/F_m_′).

With the Dual-PAM-100, P700^+^ was monitored as the absorption difference between 830 and 875 nm in a transmission mode. In analogy to the quantum yields of PSII, the quantum yields of PSI were determined using the saturation pulse method [65]. The maximum oxidizable P700, P_m_, was determined by application of the SP in the presence of far-red light at 720 nm. The decrease in P_m_ is an indicator of PSI photoinhibition. The zero P700 signal, P_o_, was determined when complete reduction of P700 was induced after the SP in the absence of far-red light. P_m_’ is the maximal P700 signal in the presence of actinic light induced by the SP. P is the oxidizable P700 signal in the presence of actinic light. The photochemical quantum yield of PSI, Y(I), was calculated as P_m_′–P. Non-photochemical energy dissipation due to the donor-side limitation, Y(ND), and due to the acceptor-side limitation, Y(NA), of PSI electron flow were determined as P–P_o_ and P_m_–P, respectively. These add up to unity with the photochemical quantum yield (i.e., Y(I) + Y(ND) + Y(NA) = 1).

For the measurements of Figure 4, Figure 5 and Figure 6, chlorophyll fluorescence was measured using a WATER-PAM (Walz, Effeltrich, Germany).

### 4.5. Neutral Red Uptake Assay

The neutral red uptake assay provides an estimation of viable cells in a solution and was carried out according to [66,67] with modifications. Briefly, marimo filamentous cells were added to each well of a 24-well plate. Four samples were prepared at 2 °C: non-damaged cells incubated in the dark as a control, photoinhibited cells after five cycles of modulated light (maximum at 500 μmol m^−2^ s^−1^ at 6 h in the light, light/dark = 12 h/12 h; see Figure 5a), and repair light-treated cells after seven and fourteen cycles of the modulated light (maximum at 250 μmol m^−2^ s^−1^ at 6 h in the light, light/dark = 12 h/12 h). These samples were incubated in a medium solution including 0.01% neutral red (3-amino-7-dimethylamino-2-methyl-phenazine hydrochloride; Sigma, cat. no. N4638) (pH 7.5), at 20 °C in the low light of 50 μmol m^−2^ s^−1^ for 48 h. Then, the cells were washed and observed with a microscope (BX53; Olympus, Tokyo, Japan). When the cell died, the dye could not be taken up; as a result, dead cells remained unstained. To confirm this, we examined the assay by exposing the filamentous cells to the incubation at 40 °C (Appendix A).

## 5. Conclusions and Future Remarks

Lake Akan in Hokkaido, Japan, constitutes an amazing natural landscape and valuable biological resources, and the spherical marimo of *A. linnaei* grown under the lake is one of the species to be preserved. However, global environmental changes could cause the ice on the lake to be more fragile in the future, indicating that marimo face a risk of exposure to strong sunlight without a thick ice layer at low temperatures, where photosynthesis is markedly suppressed. In this study, we examined the effects of the chilling and high-light stress on photoinhibition of PSII and PSI in the dissected filamentous cells from marimo. PSII was easily photoinhibited, while PSI might have been protected due to the PSII photoinhibition. We demonstrated that damaged PSII was rapidly repaired by the moderate light for 30 min at 2 °C. Further, we indicated that prolonged high illumination caused cell death of marimo. These results indicated that there was a threshold between activation and suppression of the PSII repair. The mechanism for cold adaptation of the PSII repair was further investigated. Our results in this study also hold a warning for the conservation of marimo.

## Figures and Tables

**Figure 1 ijms-24-00060-f001:**
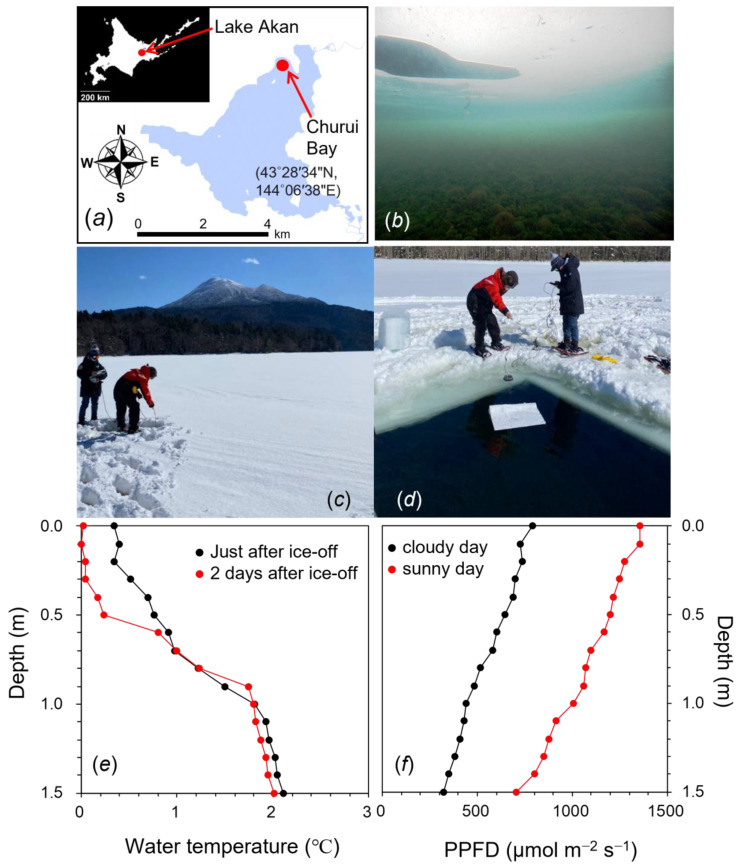
(**a**) Sampling location in Lake Akan, Hokkaido, Japan. (**b**) Marimo in Lake Akan. (**c**) Ice on the surface of Lake Akan in the winter. (**d**) Cutting out a part of the ice of the lake. (**e**) Vertical profile of the water temperature just after the ice-off (black) and two days after the ice-off (red). (**f**) Vertical profile of PPFD under the lake without the ice on cloudy (black) and sunny days (red). Measurements were conducted on a clear day in March. Ice thickness was about 50 cm.

**Figure 2 ijms-24-00060-f002:**
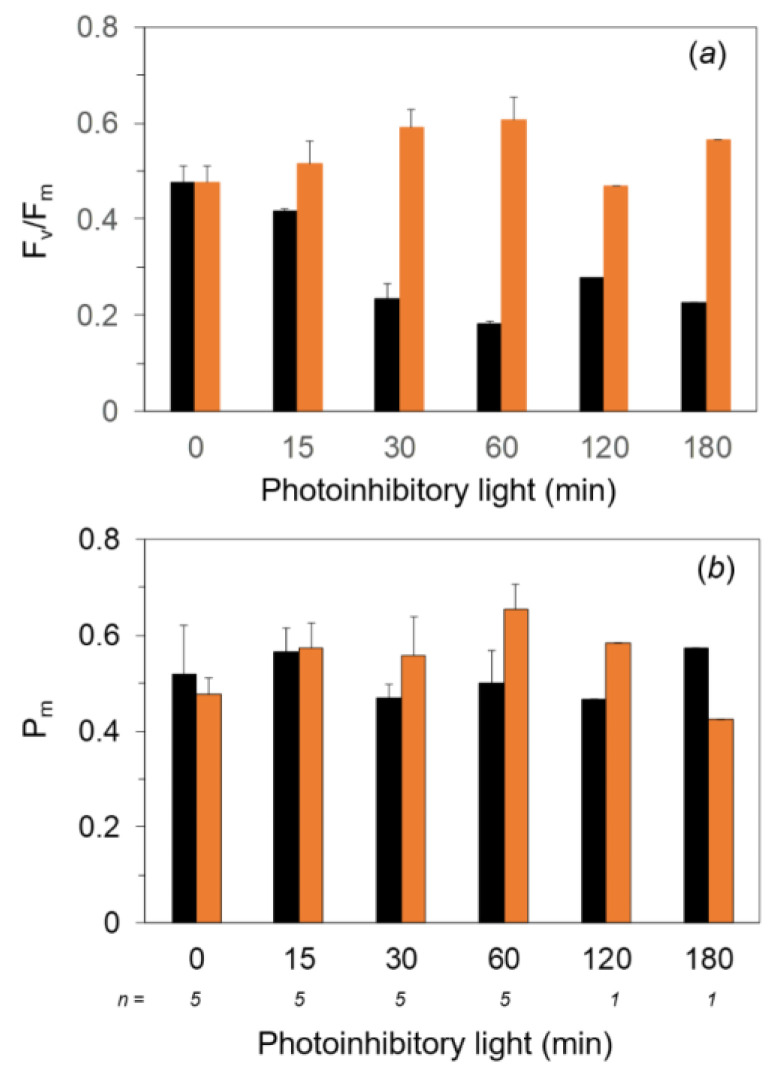
F_v_/F_m_ (**a**) and P_m_ (**b**) after the exposure to strong light and the subsequent moderate light at low temperatures in the filamentous cells dissected from the spherical marimo. Dark-treated filamentous cells were exposed to the white light at 800 μmol m^−2^ s^−1^ for up to 180 min at 2 °C on ice. After the high light for each given time, the filamentous cells were dark-treated for 30 min, and F_v_/F_m_ and P_m_ were measured at 4 °C (black bar). The photoinhibited filamentous cells at each illuminating time were irradiated by the white light at 250 μmol m^−2^ s^−1^ used as the repair light for 30 min at 2 °C on ice, and then F_v_/F_m_ and P_m_ were measured at 4 °C after the dark treatment for 30 min (orange bar). Measurements were taken in the cold room (40 Pa CO_2_, 21 kPa O_2_, at 4 °C). The values represent the means ± SD (*n* = 1–5).

**Figure 3 ijms-24-00060-f003:**
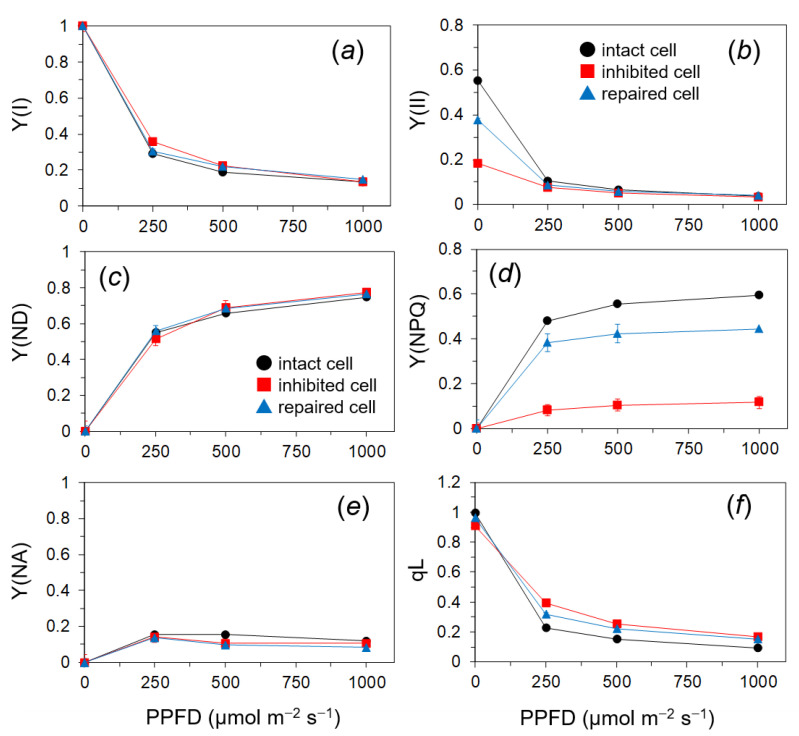
Responses of PSI (**a**,**c**,**e**) and PSII (**b**,**d**,**f**) parameters in the marimo filamentous cells to continuous light at various irradiances of actinic light in the different states of PSII: the intact cell (black line), the inhibited cell (photoinhibitory treatment for 60 min; red line), and the repaired cell (blue line). For the PSI parameter, Y(I) (**a**), Y(ND) (**c**), and Y(NA) (**e**) are shown. For the fluorescence parameters, Y(II) (**b**), Y(NPQ) (**d**), and qL (**f**) are shown. These parameters were obtained in the steady state attained at 5–10 min after the changes in the irradiance. Measurements were conducted in different filamentous cells from those used in in Figure 2. Measurements were conducted in the cold room (40 Pa CO_2_, 21 kPa O_2_, at 4 °C). The values represent the means ± SD (*n* = 4).

**Figure 4 ijms-24-00060-f004:**
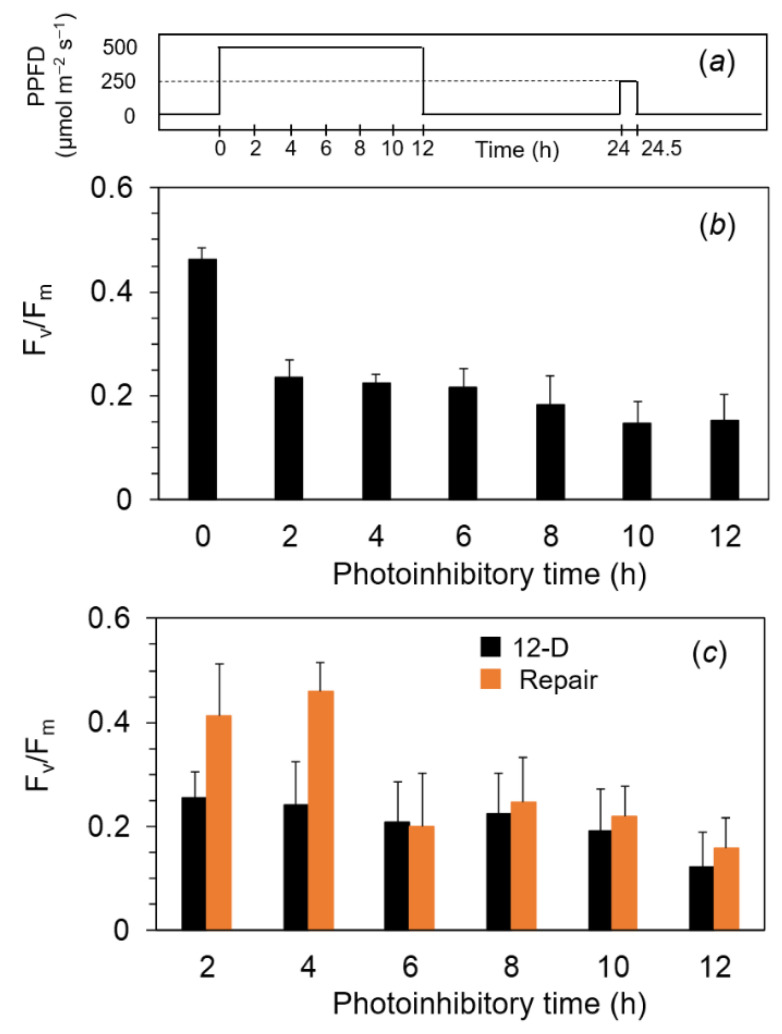
F_v_/F_m_ after the exposure to continuous high light at 500 μmol m^−2^ s^−1^ for 12 h and the subsequent moderate light at 250 μmol m^−2^ s^−1^ for 30 min at 2 °C in the marimo filamentous cells. Dark-treated filamentous cells were exposed to white light at 500 μmol m^−2^ s^−1^ for up to 12 h at 2 °C on ice (**a**). After the high light for each given time, F_v_/F_m_ in each cell was measured at 4 °C after the dark treatment for 30 min (**b**) and 12 h (black bar in c). The cells after the 12 h dark treatment were irradiated by the white light at 250 μmol m^−2^ s^−1^ used as the repair light for 30 min at 2 °C on ice, and then F_v_/F_m_ was measured after the dark treatment for 30 min (**c**) (orange bar in c). Measurements were taken in the cold room (40 Pa CO_2_, 21 kPa O_2_, at 4 °C). The values represent the means ± SD (*n* = 6–8).

**Figure 5 ijms-24-00060-f005:**
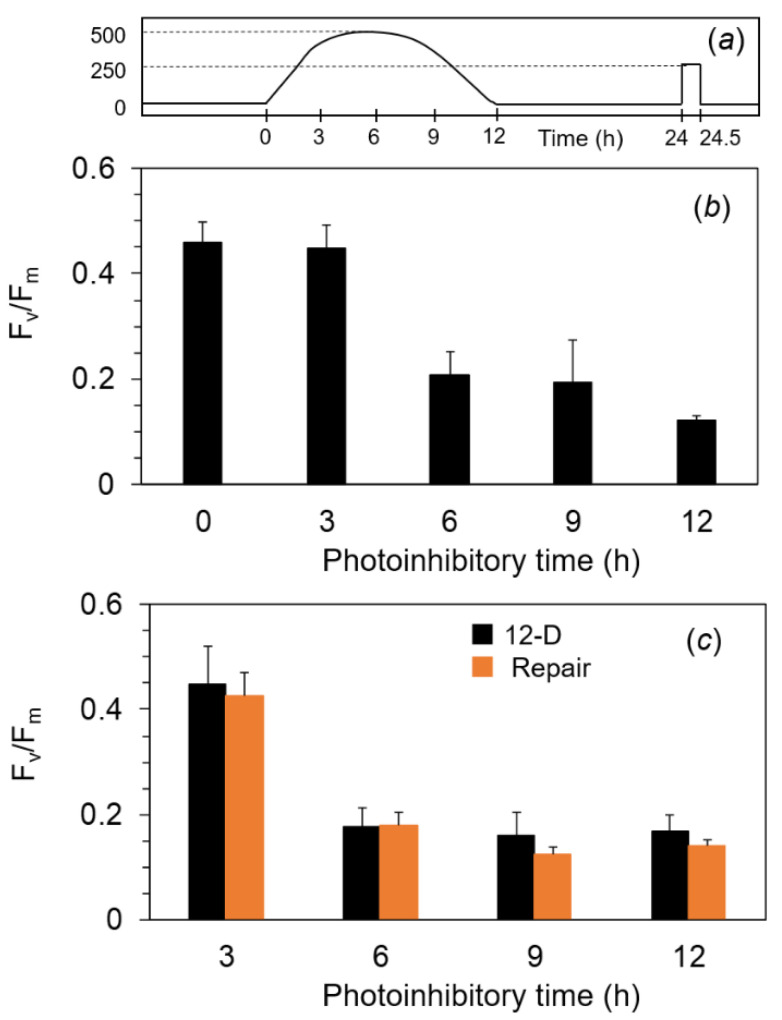
F_v_/F_m_ in the filamentous cells of marimo after the exposure to the simulated diurnal light and the subsequent moderate light at 250 μmol m^−2^ s^−1^ for 30 min at 2 °C. Dark-treated cells were exposed to the diurnal light for 12 h, which was a maximum of 500 μmol m^−2^ s^−1^ at 6 h from the onset, through the white LEDs at 2 °C on ice (**a**). After the light treatment for each given time (3, 6, and 9 h), F_v_/F_m_ was measured at 4 °C after the dark treatment for 30 min (**b**) and 12 h (black bar in **c**). The cells treated with the 12 h dark treatment were irradiated by the white light at 250 μmol m^−2^ s^−1^ used as the repair light for 30 min at 2 °C on ice, and then F_v_/F_m_ was measured after the dark treatment for 30 min (**c**) (orange bar in **c**). Measurements were taken in the cold room (40 Pa CO_2_, 21 kPa O_2_, at 4 °C). The values represent the means ± SD (*n* = 6–8).

**Figure 6 ijms-24-00060-f006:**
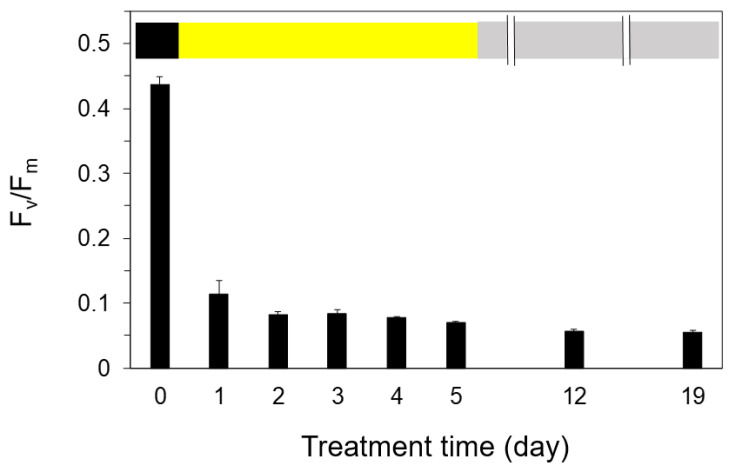
Effect of cumulative diurnal light–dark cycles on F_v_/F_m_ at 2 °C. The exposure of the filamentous cells to sine-like modulated light (maximal PPFD of 500 μmol m^−2^ s^−1^ at 6 h from the onset of the illumination) for 12 h with the dark for 12 h was repeated for the first 5 days (yellow bar). After then, the mild modulated light (maximal PPFD of at 250 μmol m^−2^ s^−1^ at 6 h from the onset of the illumination) for 12 h with the dark for 12 h was repeated for 14 days (gray bar). Measurements were taken in the cold room (40 Pa CO_2_, 21 kPa O_2_, at 4 °C). The values represent the means ± SD (*n* = 3–6).

**Figure 7 ijms-24-00060-f007:**
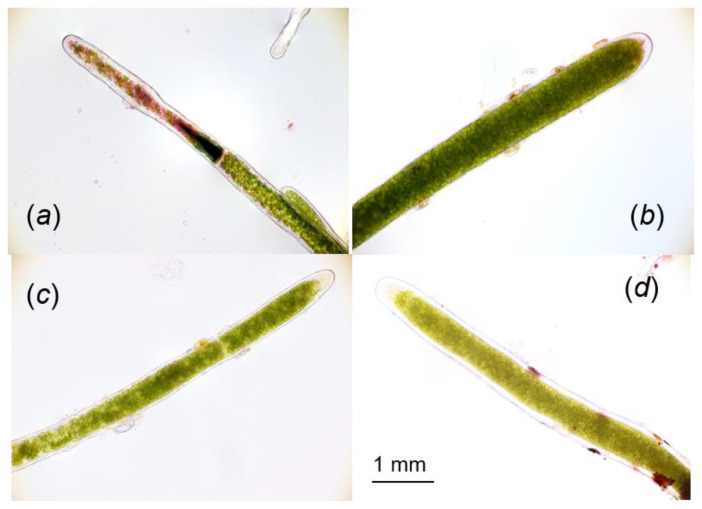
Filamentous cells of marimo used in Figure 6 with neutral red uptake assay. (**a**) Non-damaged cells incubated in the dark as a control. (**b**) Photoinhibited cells after five days of the high modulated light. (**c**) Cells treated with the repair light for 7 days after the high light for 5 days and (**d**) 14 days.

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
