# Peer review of "Effects of High Irradiance and Low Water Temperature on Photoinhibition and Repair of Photosystems in Marimo (Aegagropila linnaei) in Lake Akan, Japan"

_ijms, 2022, doi:10.3390/ijms24010060_

Round 1

Reviewer 1 Report

Abstract:

It is simple, Missing information. You can better present the results achieved at work

Introduction:

A better description of photoinhibition/photooxidation is recommended. And about the repair systems.

Metodology and Results:

The conditions of keeping the filaments in lake water under controlled conditions are misleading, they do not have knowledge of factors associated with nitrients, pH and O2/CO2 ratio. All of these factors directly alter PSII. They had to control for all factors, such as using a synthetic culture medium. 

The field data directly from the Lake are punctual for the day of sampling, they are not conclusive, several measurements would be needed at different times of the year. I would recommend removing them from the document.

They must expand the description of the experiment for measurements with the fluorometer or attach a diagram or photograph. Doubts arise, how were the experimental units isolated in a 96-well plate?

Discussion

Comparisons with vascular plants cannot be made, they are not valid. I recommend focusing the discussion solely on algae

By using dissected filaments, the results are only valid at the cellular level, it cannot be matched to the algae in the lake because they did not take into account the morphological structure of the colony (sphere), that also provides protection against high radiation. They should mean that the results give an indication or clue of what may happen to the algae in the lake.

Author Response

In the revised manuscript, the parts which we revised in response to the comments are in red. We also modified some parts to improve the paper. These parts are also in red. In this letter, the comments by the reviewers are in Arial italic, while our responses are in Century.

Reviewer 1

Abstract:

It is simple, Missing information. You can better present the results achieved at work

>> Thank you for helpful comment. We entirely re-wrote the Abstract, and in the revised version, concentrated on the novel findings of the present study in addition to brief introduction, aim and results.

Introduction:

A better description of photoinhibition/photooxidation is recommended. And about the repair systems.

>> We’re sorry for poor information about photoinhibition and repair systems. In the revised version, we briefly explained the photosynthetic electron transport, and the description of photoinhibition and repair system was followed so that the readers can easily understand the story line of this part. Part of ‘photoinhibition’ was slightly modified, and that of ‘repair system’ was re-written; we added the description about the outline of the repair system and the requirement of the photons to repair PSII. See lines 52 – 75.

Metodology and Results:

The conditions of keeping the filaments in lake water under controlled conditions are misleading, they do not have knowledge of factors associated with nitrients, pH and O2/CO2 ratio. All of these factors directly alter PSII. They had to control for all factors, such as using a synthetic culture medium.

>> Thank you for your comment. In the present study, the algae were maintained in lake water in sufficient volume for the period before the experiment, and the experiment was also conducted in lake water. The condition was not controlled during the photoinhibitory treatments. The aim of our experiment was to replicate injury in a natural environment. We did not attempt to study the characteristics of cells cultured as experimental organisms. For this reason, we conducted the experiment in the lake water in the habitat area immediately after collecting the algae. Lake Akan is a volcanic lake in a caldera and has special lake water. We believe it is difficult to reproduce this with artificial culture media. As this reviewer pointed out, the factors such as nutrients, pH, O2/CO2 ratio in the medium might affect the results, but we do not know how these conditions would affect the results. The photoinhibitory treatment at 20℃ for 5 days showed no PSII photoinhibition as shown in Supplementary Fig. S5. However, the lake water per se might have caused large effects at low temperatures. Because we agree that the experiment should be done in the controlled condition, we will test the composition of the culture medium suitable for Marimo culture and experiments.

The field data directly from the Lake are punctual for the day of sampling, they are not conclusive, several measurements would be needed at different times of the year. I would recommend removing them from the document.

>> Thank you for your comment. As this reviewer pointed out, the field data in Lake Akan was not continuously measured, and thus might not be conclusive. However, the data of Fig. 1 are quite important, because we have carried out the following experiments based on these data. So, we cannot remove Fig. 1 or the related part in the manuscript. We agree with your comment that the measurement should be done successively during winter. We are planning to measure the light and temperature in Lake Akan from this winter to next spring. However, the data from the nearest meteorological station would support that our field data and experimental conditions are reasonable.

They must expand the description of the experiment for measurements with the fluorometer or attach a diagram or photograph. Doubts arise, how were the experimental units isolated in a 96-well plate?

>> We’re sorry for the fact that there was a mistake in the original version. We wrote “96-well plate” by mistake. We meant “24-well plate”. In the revised version, we added the detailed explanation for the experimental procedures of photoinhibition and repair, and provided a new supplementary Fig. S6, which includes the image of picking the filamentous cells of A. linnaei up from the surface parts of the spherical Marimo with tweezers, and the image of the photoinhibitory and repair treatments. See lines 341 – 355 in the revised version.

Discussion

Comparisons with vascular plants cannot be made, they are not valid. I recommend focusing the discussion solely on algae

>> Thank you for your comment. Concerning the response of PSI to the fluctuating light (in lines 249 – 259), plants and algae have shown similar data in that PSI is sensitive to the fluctuating light and that its protection needs the photosynthetic alternative electron pathways including cyclic electron flows. Therefore, these descriptions and other associated descriptions about cyclic electron flows have been kept in the discussion section.

              (In lines 264 – 267 of the previous manuscript,) we thought that the description about the sustained NPQ in overwintering plants cannot be compared to algae probably. Thus, we omitted these sentences.

By using dissected filaments, the results are only valid at the cellular level, it cannot be matched to the algae in the lake because they did not take into account the morphological structure of the colony (sphere), that also provides protection against high radiation. They should mean that the results give an indication or clue of what may happen to the algae in the lake.

>> We would like to thank your helpful comment. We agree with you. In the last paragraph of the discussion section in the revised version, we added the description “In the present study, as we used the dissected filamentous cells, the results might be only valid at the cellular level”. On that basis, we modified this paragraph.

Reviewer 2 Report

This is an interesting manuscript. Authors examined photoinhibition of PSII and PSI in Marimo against the exposure to high light at 2℃, and recovery at 2℃ from the photoinhibited state, in Marimo grown in Lake Akan in winter. The manuscript is mostly well-written and contains relevant information. I have checked carefully the revised version and have few suggestions to improve the text.

1.Introduction---Is Aegagropila linnaei endemic to Japan? Please add a brief description of the taxonomic status of this species as well as its habitat, Lake Akan.

2.---Figure 1. (a)--- Add latitude and longitude to the map.

3. Minor errors:

line 25---“light.-To”, “-” is redundant.

line 172--- Replace “hours” with “h”.

line 310---“10cm”, Add a space before the unit.

Author Response

In the revised manuscript, the parts which we revised in response to the comments are in red. We also modified some parts to improve the paper. These parts are also in red. In this letter, the comments by the reviewers are in Arial italic, while our responses are in Century.

Reviewer 2

This is an interesting manuscript. Authors examined photoinhibition of PSII and PSI in Marimo against the exposure to high light at 2℃, and recovery at 2℃ from the photoinhibited state, in Marimo grown in Lake Akan in winter. The manuscript is mostly well-written and contains relevant information. I have checked carefully the revised version and have few suggestions to improve the text.

1.Introduction---Is Aegagropila linnaei endemic to Japan? Please add a brief description of the taxonomic status of this species as well as its habitat, Lake Akan.

>> Thank you for your question. Aegagropila linnaei is a freshwater green alga of the order Cladophorales and is distributed not only in several lakes in Japan but also in quite a few lakes of some regions in Europe, East Asia and North America. Lake Akan is the only lake in which spherical ‘Marimo’ is found in Japan, and further the only lake in which giant Marimo in the 30-cm diameter is found in the world. A. linnaei has a branched filamentous form and grows as attached, floating unattached, and aggregated growth forms. The aggregate often forms beautiful spherical shapes known as “lake balls” or “Marimo” in Japanese. Marimo grown in Lake Akan has an experience in dynamic changes in water temperatures through the year.  In summer of Lake Akan, water is 20 – 27℃ in temperature, but water temperature lowers to 1 – 4℃ in winter.

In the present study, we focused on the effect of the low temperature on photosynthesis in Marimo against the exposure to the high illumination. As the reviewer suggests, we added a brief explanation of the taxonomic status and the habitat of Marimo in Lake Akan, in the revised manuscript. See lines 31 – 51.

2.---Figure 1. (a)--- Add latitude and longitude to the map.

>> Thank you for a helpful comment. We added latitude and longitude to the map in Fig, 1a.

  1. Minor errors:

line 25---“light.-To”, “-” is redundant.

line 172--- Replace “hours” with “h”.

line 310---“10cm”, Add a space before the unit.

>> Thank you for correcting the errors. We have done.

Round 2

Reviewer 1 Report

The authors attach the requested information, and make the pertinent clarifications regarding the lack of control of some experiments. According to the first review.

A more complete review of references on chlorophyll fluorescence in algae is still lacking, because they continue to use examples from vascular plants.

Author Response

In the revised manuscript, the parts which we revised in response to the comments are in red. We also modified some parts to improve the paper. These parts are also in red. In this letter, the comments by the reviewers are in Arial italic, while our responses are in Century.

Reviewer 1

The authors attach the requested information, and make the pertinent clarifications regarding the lack of control of some experiments. According to the first review.

A more complete review of references on chlorophyll fluorescence in algae is still lacking, because they continue to use examples from vascular plants.

>> We are sorry for poor references for chlorophyll fluorescence in algae. In the R2-revised version, we cited the reviews about PAM measurement on algal research in the Materials and Methods section and some papers focused on algae in the Introduction and Discussion sections.